# Optimization for the Process Parameters of Nickel–Titanium Nitride Composites Fabricated via Jet Pulse Electrodeposition

**DOI:** 10.3390/nano14242034

**Published:** 2024-12-18

**Authors:** Xue Guo, Dehao Tian, Chaoyu Li, Xiang Li, Wei Li, Mengyu Cao, Fengwu Zhang, Baojin Wang

**Affiliations:** 1College of Mechanical Science and Engineering, Northeast Petroleum University, Daqing 163318, China; gx_cmy@nepu.edu.cn (X.G.); 238003040822@stu.nepu.edu.cn (D.T.); 208001040066@stu.nepu.edu.cn (B.W.); 2School of Mechanical and Electrical Engineering, Sanming University, Sanming 365004, China; 20230368@fjsmu.edu.cn; 3College of Civil Engineering, Dalian Minzu University, Dalian 116600, China; lixiang@dlnu.edu.cn; 4College of Petroleum Engineering, Northeast Petroleum University, Daqing 163318, China

**Keywords:** Ni/TiN composites, jet pulse electrodeposition, corrosion resistance, RSM design, parameters optimization

## Abstract

The corrosion resistance of nickel–titanium nitride (Ni/TiN) composites is significantly influenced by the operation parameters during the jet pulse electrodeposition (JPE) process. The effect of current density, jet rate, TiN concentration, and duty cycle impact on the anti-corrosion property of Ni/TiN composites were investigated and optimized using the response surface method (RSM). After the optimization of the operation parameters, the corrosion current of Ni/TiN composites decreased from 9.52 × 10^−5^ A/cm^2^ to 4.63 × 10^−5^ A/cm^2^. The corrosion current of Ni/TiN composites decreased initially and then increased with an increase in current density, jet rate, TiN concentration, and duty cycle. During the jet electrodeposition process, the influence of the duty cycle on the corrosion current of Ni/TiN composites was comparatively insignificant, whereas the concentration of TiN had a significant effect on the corrosion current. The error rate between the predicted value and the measured result from the corrosion current of Ni/TiN composites was only 0.64%, indicating the high accuracy of fitting the model. Furthermore, X-ray diffraction (XRD) patterns and scanning electron microscope (SEM) images revealed that the optimized Ni/TiN composites comprised significant Ti content, fine nickel gain, and a compact, smooth structure. In addition, the electrochemical measured results demonstrated that the optimized Ni/TiN composites possessed a low self-corrosion current and high self-corrosion potential. These findings show that the optimized composites have a substantially greater corrosion resistance compared to two other unoptimized Ni/TiN composites.

## 1. Introduction

Researchers from both industry and academia have shown considerable interest in nickel-matrix composites (NMCs) in recent years due to their exceptional physical properties, chemical stability, and cost-effectiveness [1,2,3]. Although their primary function is to prevent corrosion of the metal matrix, NMCs are widely used in the transportation, mechanical, and processing industries. For instance, Zhang et al. [4] prepared electrodeposited Ni-W/TiN films to protect X52 steel used in petroleum and natural gas applications. Liang et al. [5] fabricated Ni/SrSO_4_ composite coatings to enhance the tribological performance of bearings. Furthermore, the various reinforced phases are introduced into nickel-matrices. Zhang et al. [6] produced Ni/SiC coatings to improve the abrasion resistance of the Q235 steel substrate. Ma et al. [7] investigated the anti-corrosion ability of Ni/Al_2_O_3_ composites prepared at various ultrasonic powers. Ge et al. [8] studied the corrosion resistance of pure Ni and Ni/ZrO_2_ coatings manufactured at different annealed temperatures.

Compared to traditional electrodeposition, the jet pulse electrodeposition (JPE) technique using a circular electrolyte sprayed and deposited onto a substrate surface could improve the microstructure, corrosion resistance, and wear resistance of the coatings [9]. Therefore, JPE offers significant advantages, including a high efficiency, broad adaptability, and cost-effectiveness, making it the preferred choice for metal matrix coatings [10,11]. For instance, Fan et al. [12] prepared a nano-crystalline copper electrode material on a stainless steel substrate. Zhao et al. [13] studied the influence of pulse parameters on the microstructure and anti-corrosion performance of Cu/Al_2_O_3_ composites. The influence of various surfactant types on the structure and anti-corrosion property of electrodeposited Cu/TiO_2_ composites was examined by Ning et al. [14].

Currently, researchers are actively involved in the synthesis and characterization of NMCs, resulting in a wealth of research achievements. For instance, Cao et al. [15] produced the ultrasonic-assisted jet electrodeposited Ni/SiC composite coatings. They demonstrated that the SiC concentration had an obvious influence on the anti-corrosion performance and structure of Ni/SiC composites. Jiang et al. [16] fabricated Co-Ni/SiO_2_ composites on a copper substrate through the JPE technique to identify the feasible pulse current density capable of achieving high-quality composites. Li et al. [17] prepared Ni-Co/BN(h) composites and found that the Ni-Co/BN(h) composites deposited at a pulse frequency (4 kHz) and a duty cycle (0.7) possessed the highest content of Co and BN(h). Previous studies [18,19,20] have demonstrated the significant effect of a single factor (i.e., current density, electrolyte jet rate, and TiN concentration) on the corrosion resistance of Ni/TiN composites. Furthermore, current studies focus on the orthogonal experiment for analyzing and optimizing the operational parameters of NMCs. However, the accuracy of the optimal model is not high due to the discrete nature of the orthogonal experiment. The response surface method (RSM) approach offers several benefits over the orthogonal experiment method, including enhanced precision, visual analysis, and model continuity [21,22]. These advantages render it suitable for evaluating and optimizing the anti-corrosion performance of NMCs produced using the JPE technique. Furthermore, there is a scarcity of reports examining the relationship between various operational parameters and the corrosion resistance of NMCs developed using the JPE method. Hence, the relationship between operational parameters and the anti-corrosion properties of Ni/TiN composites was, therefore, investigated and optimized in this study using the RSM approach. Furthermore, X-ray diffraction (XRD) and scanning electron microscopy (SEM) with energy dispersive spectroscopy (EDS) were employed to verify the operational parameters for making Ni/TiN composites after the optimization of the RSM method.

## 2. Experiment

### 2.1. Coatings Preparation

The cathode used in this study was a Q235 steel substrate measuring 30 mm × 30 mm × 5 mm. The anode, a nickel nozzle with a diameter of 4 mm, was employed. Abrasive paper of 400, 800, and 1200 grits was used to polish the cathode before JPE. Furthermore, the rust remover solution contained 150 g/L H_2_SO_4_, 300 g/L HCl, and 5 g/L OP(Octylphenol ethoxylate). Also, the oil-remover solution included 80 g/L NaOH and 50 g/L Na_3_PO_4_. The polished sample was then rinsed with distilled water and activated for 20 s in the mixed solution of 0.05 g/L sodium dodecyl sulfate and 12 wt.% HCl. Finally, the sample was rewashed with distilled water. Figure 1 illustrates the schematic JPE system employed for the development of Ni/TiN composites. The JPE system comprised two main units: the circulating unit comprising a circulating pump, electrolytic recycling tank, servo system, and flow-meter, and the electrodeposition unit consisting of pulse power, Q235 steel substrate, nickel nozzle, bench, and electrolytic bath. The space between the anode and cathode was adjusted to 15 mm. The composition of the electrolyte is detailed in Table 1. All chemical reagents used in the JPE process were of analytical grade and required no additional purification. NiSO₄ and NiCl_2_ provided Ni^2+^ ions for the deposition of coatings. Meanwhile, NiCl_2_ provide Cl^−^ ions to prevent the passivation of the anode. C_6_H_8_O_7_ (citric acid) served as the coordination agent, while H_3_BO_3_ acted as a buffering agent in the electrolyte to stabilize the pH during the deposition process. The cetyltrimethyl ammonium bromide (CTAB) functioned as a cationic surface active agent to decrease the surface energy of TiN particles, facilitate their transport towards the negative charged electrode and incorporation into the metal layer on the cathode.

### 2.2. Statistical Design of Experiment

The central composite design (CCD) of four pilot factors was used to optimize the corrosion resistance of Ni/TiN composites, contributing to the remarkable accuracy of fitting curves [23]. The number of each pilot factor was calculated by Equation (1).
(1)$$X_{i}=\frac{x_{i}-x_{0}}{\Delta x_{i}}$$
where Xi represents the number of independent variables, xi denotes the actual value of the independent variable, x_0_ is the center value of the independent variable, and ∆xi presents the variable step length of the independent variable. Furthermore, the corrosion current (Y) serves as a response value and is arranged to the minimal value. A variety of independent variables, including current density (X_1_), jet rate (X_2_), TiN concentration (X_3_), and duty cycle (X_4_), contribute to the variation of corrosion current. These variables are closely related to the microstructure and corrosion resistance of composites. The CCD procedure also included the development of four pilot factors as continuous values. According to our experience, the range of current density, jet rate, TiN concentration, and duty cycle (the percentage of turn-on time of current during each pulse period) were selected at 20~60 A/dm^2^, 1.2~2.4 m/s, 5~15 g/L, and 30~60%, respectively. The specific pilot factors and levels are displayed in Table 2. The experimental findings of the corrosion current of Ni/TiN composites are presented in Table 3. Furthermore, the polynomial model equation for the corrosion current of Ni/TiN composites was calculated using Equation (2), which was based on the least square method. Moreover, the variance analysis of the fitting model is listed in Table 4.
(2)$$\begin{array}{l} Y=4.666-0.0856\;\times \;X_{1}-0.117\;\times \;X_{2}-0.064\;\times \;X_{3}-0.092\;\times \;X_{4}-0.188\;\times \;X_{1}X_{2}+0.342\;\times \;X_{1}X_{3}+\\ 0.198\;\times \;X_{2}X_{3}+0.347\;\times \;X_{1}X_{4}+0.371\;\times \;X_{2}X_{4}-0.064\;\times \;X_{3}X_{4}+0.824\;\times \;X_{1}^{2}+0.974\;\times \;X_{2}^{2}+\\ 0.574\;\times \;X_{3}^{2}+0.979\;\times \;X_{4}^{2} \end{array}$$

### 2.3. Characterization

The X-ray diffractometer (XRD, D5000, Siemens, Munich, Germany) was used to determine the phase structure of Ni/TiN composites. The operational parameters included a scanning range of 20° to 90°, Cu Kα radiation employed as an X-ray source, a 0.02°/s scanning rate, and an applied power of 20 kV. The scanning electron microscope (SEM, S3400, HITACHI, Tokyo, Japan) equipped with an energy dispersive spectrometer (EDS, INCA X-Max-20, Oxford, UK) was used to observe the surface morphology and elemental content of Ni/TiN composites before and after optimization. The CS-350 style electrochemical workstation was used to ascertain the corrosion current of Ni/TiN composites in a 5 wt.% NaCl solution. The working electrode, reference electrode, and auxiliary electrode were Ni/TiN composites, saturated calomel electrode (SCE), and platinum electrode, respectively. The tested area of Ni/TiN composites immersed in 5 wt.% NaCl solution were 1 cm^2^, which were measured at a frequency of 2 Hz, scanning speed of 1 mV/s, and indoor temperature for 10 min.

## 3. Results and Discussion

### 3.1. Effect of Current Density and Jet Rate on the Corrosion Resistance

Figure 2 depicts the response surface relationship between the corrosion current of Ni/TiN composites and the current density and jet rate. The lowest corrosion current observed in Ni/TiN composites was 4.68 × 10^−5^ A/cm^2^, achieved at a jet rate of 1.81 m/s and a current density of 39.85 A/dm^2^. Moreover, with increasing current density and jet rate, the corrosion current of Ni/TiN composites showed an initial decrease and then a subsequent increase. The following factors may provide explanations for these results: (1) the increase in current density generated large cathode polarization and promoted the co-deposition of TiN particles and Ni^2+^ ions, leading to the grain size being refined, an increase in the TiN content, and a decrease in the corrosion current of Ni/TiN composites [24]. However, the substantial current density triggered considerable hydrogen evolution and concentration polarization, ultimately resulting in the development of surface defects in the coatings, including pits and fractures. Hence, the corrosion current of Ni/TiN composites increased [25]. Furthermore, the high current density formed an uneven and coarse surface, resulting in an increase in the corrosion current of Ni/TiN composites. (2) The increase in jet rate could move a larger amount of TiN particles to the cathode, which improved the content of TiN particles in the Ni/TiN composites [26]. Alternatively, an appropriate jet rate could reduce the thickness of the diffusion layer and inhibit the rapid growth of the nickel grain, leading to the formation of a dense and flatted microstructure and a reduction in the corrosion current [27]. These results are consistent with the research of Zhang et al. [28], who reported on the effect of current density and jet rate on the corrosion resistance of NMCs.

### 3.2. Effect of Current Density and TiN Concentration on the Corrosion Current

Figure 3 displays the response surface relationship between the corrosion current of Ni/TiN composites and the current density and TiN concentration. The lowest corrosion current for Ni/TiN composites (4.69 × 10^−5^ A/cm^2^) was achieved at a current density of 39.96 A/dm^2^ and a TiN concentration of 10.57 g/L. Furthermore, as both the TiN concentration and current density increased, the corrosion current of Ni/TiN composites initially decreased and subsequently increased. These results can be attributed to the same factors previously discussed regarding the influence of current density on the corrosion current of Ni/TiN composites. Furthermore, the insufficient TiN concentration could not provide enough nucleation points, and it limits the rapid growth of Ni grains. Therefore, this led to the development of a coarse and uneven structure and the generation of a high corrosion current in the Ni/TiN composites [29]. However, higher concentrations of TiN contributed to significant concentration polarization and a high viscosity of the electrolyte, leading to the coarse and uneven microstructure generated. The finding is proven by the conclusion of Liu et al. [30], who reported on the effect of TiN concentration on the microstructure of Ni-W/TiN coatings. These factors hindered TiN particles embedded into the nickel matrix and showed a pronounced increase in the corrosion current of Ni/TiN composites. The conclusion is similar to the research conducted by Li et al. [31] regarding the influence of reinforced phase particles on the anti-corrosion ability of Ni/BN/TiC layers.

### 3.3. Effect of Current Density and Duty Cycle on the Corrosion Current

Figure 4 illustrates the response surface relationship between the corrosion current of Ni/TiN composites and the duty cycle and current density. Achieving the minimum corrosion current of 4.7 × 10^−5^ A/cm^2^ for Ni/TiN composites required a current density of 41.26 A/dm^2^ and a duty cycle of 0.45. Furthermore, as both the duty cycle and current density increased, the corrosion current of Ni/TiN composites initially decreased and then increased. Similar to the previous analysis, the reasons for the observed result can be attributed to the following: (1) the corrosion current of Ni/TiN composites can be affected by the current density, as previously explained. (2) The duty cycle had a significant effect on the structure and corrosion resistance of NMCs, according to Yu et al. [32]. The prolonged duration of the pulse interval caused by the low duty cycle was detrimental to TiN particles embedded into the nickel matrix and the formation of a compact and smooth structure, which reduced the anti-corrosion ability of the composites in contact with the corrosion liquid and increased the corrosion current of Ni/TiN composites [33]. (3) However, the higher duty cycle resulted in the growth of the average current density and produced a significant hydrogen evolution, which eventually decreased the anti-corrosion performance and increased the corrosion current of the Ni/TiN composites [34].

### 3.4. Effect of Jet Rate and TiN Concentration on the Corrosion Current

Figure 5 illustrates the response surface relationship between the corrosion current of Ni/TiN composites and the jet rate and TiN concentration. The lowest corrosion current of 4.68 × 10^−5^ A/cm^2^ for the Ni/TiN composites was achieved at a jet rate of 1.86 m/s and a TiN concentration of 10.49 g/L. As the jet rate and TiN concentration increased, the corrosion current of the Ni/TiN composites showed an initial decrease followed by a subsequent increase. The same reasons for the influence of jet rate and TiN concentration on the corrosion current of the Ni/TiN composites are discussed in the previous chapter.

### 3.5. Effect of Jet Rate and Duty Cycle on the Corrosion Current

Figure 6 shows the response surface relationship between the corrosion current of Ni/TiN composites and jet rate and duty cycle. The lowest corrosion current of 4.67 × 10^−5^ A/cm^2^ for Ni/TiN composites was achieved at a jet rate of 1.85 m/s and a duty cycle of 0.45. Moreover, the corrosion current of Ni/TiN composites initially decreased and then increased with an increase in both the jet rate and duty cycle. The same reasons for the effect of the jet rate and duty cycle on the corrosion current of the Ni/TiN composites are discussed in detail in Section 3.1 and Section 3.3.

### 3.6. Effect of TiN Concentration and Duty Cycle on the Corrosion Current

The response surface relationship between the duty cycle and TiN concentration for the corrosion current of Ni/TiN composites is illustrated in Figure 7. The Ni/TiN composites constructed with a duty cycle of 0.45 and a TiN concentration of 10.26 g/L showed the lowest corrosion current, measuring 4.69 × 10^−5^ A/cm^2^. Furthermore, the corrosion current of Ni/TiN composites initially decreased followed by a subsequent increase with the increase in both the TiN concentration and duty cycle. The attributes for the influence of TiN concentration and duty cycle on the corrosion current of Ni/TiN composites are discussed in detail in Section 3.2 and Section 3.3.

### 3.7. Optimization of Result and Experimental Verification

The operational parameters of Ni/TiN composites were optimized using the least square method to minimize the propensity of corrosion current to achieve the highest-possibility corrosion resistance. The pulse parameters were rounded to two decimal places to facilitate the experimental operation. The optimized conditions for minimizing corrosion current in Ni/TiN composites are depicted in Figure 8. These conditions include the following: a current density of 40.95 A/dm^2^, a jet rate of 1.83 m/s, a TiN concentration of 10.13 g/L, and a duty cycle of 0.45.

After optimization, the predicted corrosion current value was determined to be 4.66 × 10^−5^ A/cm^2^, while the measured corrosion resistance value was 4.63 × 10^−5^ A/cm^2^. The error rate between the predicted and measured corrosion currents of Ni/TiN under optimal operational parameters was only 0.64%. The results indicate that the accuracy of the corrosion current prediction for Ni/TiN composites optimized by JMP11 software with RSM was higher than that of the prediction optimized by the orthogonal experiment [35]. Moreover, the corrosion current of Ni/TiN composites significantly decreased from 9.52 × 10^−5^ to 4.63 × 10^−5^ A/cm^2^ after optimization.

For contrast, the Ni/TiN composites produced at a current density of 20 A/dm^2^, jet rate of 1.2 m/s, TiN concentration of 5 g/L, and duty cycle of 0.3 are denoted as JPE-1. The Ni/TiN obtained at a current density of 40 A/dm^2^, jet rate of 1.8 m/s, TiN concentration of 10 g/L, and duty cycle of 0.45 are recorded as JPE-2. The Ni/TiN composites prepared at a current density of 40.95 A/dm^2^, jet rate of 1.83 m/s, TiN concentration of 10.13 g/L, and duty cycle of 0.45 are regarded as JPE-3. Scanning electron micrographs of the surface morphological characteristics of Ni/TiN composites are illustrated in Figure 9. The element compositions and contents of Ni/TiN composites are displayed in Table 5. Xia et al. [36] reported that the plating parameters had a significant influence on the surface morphology of NCBs. The SEM analysis revealed that the optimization of the plating parameters significantly enhanced the nickel grain size and surface uniformity of the Ni/TiN composite coatings. A considerable aggregation of TiN nanoparticles and coarse nickel particles was observed in JPE-1, while JPE-2 showed only a slight aggregation of TiN nanoparticles and small nickel grains. However, refined nickel grains and a dense, flatted microstructure were observed in JPE-3. Furthermore, the EDS results indicated that the Ti content of JPE-3 (12.95 wt.%) was significantly higher than that of JPE-1 (4.26 wt.%) and JPE-2 (10.74 wt.%). The observed result can be ascribed to the abundance of TiN particles, which significantly enhance the refinement strength of nickel grains [37]. This conclusion aligns with the research findings obtained by Fan et al. [38], who reported on the effect of operational parameters on the microstructure of NMCs obtained via an electrodeposition technique.

Figure 10 shows the cross-sectional surface of Ni/TiN composites. The SEM pictures indicate the increase in the thickness of Ni/TiN composites after varying the process parameters at the same electrodeposition time. The thickness of JPE-1, JPE-2, and JPE-3 samples was determined to be 95.3, 102.5, and 120.6 μm, respectively. The deposition speed (s) can be calculated according to the following equation:(3)$$S=\frac{H}{t}$$
where H represents the composite thickness and t denotes the electrodeposition time.

The deposition speed of the JPE-1, JPE-2, and JPE-3 samples was approximately 1.59 μm/min, 1.71 μm/min, and 2.01 μm/min, respectively.

Figure 11 shows the XRD patterns of Ni/TiN composites. The average grain sizes of the Ni/TiN composites are listed in Table 6. The Ni grains of each of the three Ni/TiN composites presented a face-centered cubic (FCC) structure [39]. The relative Ni diffraction peaks of JPE-1 and JPE-2 were relatively sharp and high, whereas the relative Ni diffraction peak of JPE-3 appeared broad and low. According to the Scherrer formula,
(4)$$D=\frac{K\lambda }{\beta \cos\theta }$$
where D is the average grain size, K is Scherrer’s constant (0.89), λ is the X-ray wavelength (for CuKα, λ = 1.5406 Å), β is the full width at half maximum (FWHM) of the diffraction peak (in radians), and θ is the diffraction angle of the X-ray. By applying the Scherrer formula, the average sizes of Ni grains in JPE-1, JPE-2, and JPE-3 were estimated as 0.64, 0.35, and 0.12 μm, respectively. These results correspond to the nickel grain sizes of Ni/TiN composites presented in Figure 9.

The appearance of primary diffraction peaks in JPE-1, JPE-2, and JPE-3 correspond to the presence of both a Ni phase and a TiN phase. The three diffraction peaks associated with nickel appeared at 44.8°, 52.2°, and 76.8°, corresponding to the (111), (200), and (220) crystal planes, respectively. For the case of the TiN phase, the diffraction peaks were observed at 36.7°, 42.6°, and 61.8°, corresponding to the (111), (200), and (220) crystal planes, respectively. These findings are consistent with the conclusions presented by Zhu et al. [40].

Figure 12 shows the polarization curves and Nyquist plots of Ni/TiN composites. The specific data is listed in Table 7. According to Li et al. [41], coatings with low corrosion current, high corrosion potential, and large corrosion impedance were found to have superior corrosion resistance. The extrapolation method showed that corrosion potentials (*E*_corr_) of JPE-1 and JPE-2 were determined to be −0.762 V and −0.441 V, respectively, as shown in Figure 12a. In contrast, JPE-3 had the highest corrosion potential, with the value of −0.384 V. Furthermore, the corrosion currents (*I*_corr_) of JPE-1 (9.52 × 10^−5^ A/cm^2^) and JPE-2 (4.71 × 10^−5^ A/cm^2^) were higher than that of JPE-3 (4.63 × 10^−5^ A/cm^2^). JPE-3 had a higher impedance value compared to JPE-1 and JPE-2, as demonstrated in Figure 12b. Hence, these findings indicate the significantly superior corrosion resistance of JPE-3 compared with JPE-1 and JPE-2.

Figure 13 presents the SEM images and EDS results of the corroded surface of Ni/TiN composites. The SEM graphs demonstrate the presence of obvious corrosion pits on the surfaces JPE-1 and JPE-2, while slight corrosion cracks appeared on the JPE-3 surface. The EDS results indicate the presence of Fe element on the corrosion surface of JPE-1, illustrating that the Q235 steel substrate was corroded by the NaCl solution. The dense structure and high TiN concentration in JPE-3 led to the inhibition of corrosion produced by a 5 wt.% NaCl solution, as demonstrated in Figure 9 and Figure 11. The conclusion is consistent with the findings of Li et al. [42].

## 4. Conclusions

In conclusion, three Ni/TiN composites were deposited on the surface of a Q235 steel substrate. After optimization by the RSM method for operational parameters, the corrosion current of Ni/TiN composites was significantly decreased from 9.52 × 10^−5^ A/cm^2^ to 4.63 × 10^−5^ A/cm^2^. Furthermore, the difference between the predicted and measured corrosion currents of Ni/TiN composites was only 0.64%. Also, the JPE-1 sample showed coarse nickel grains and significant agglomeration of TiN nanoparticles, while the JPE-3 sample showed refined nickel grains and a uniform distribution of TiN nanoparticles. The XRD patterns confirmed the SEM findings. Furthermore, the Ti content of JPE-3 (12.95 wt.%) was significantly higher than that of JPE-1 (4.26 wt.%) and JPE-2 (10.74 wt.%). The corrosion behavior revealed that JPE-3 was significantly less susceptible to corrosion damage than JPE-1 and JPE-2 when immersed in a 5% NaCl solution. The appearance of Fe in JPE-1 suggests that the NaCl solution corroded the Q235 steel substrate.

## Figures and Tables

**Figure 1 nanomaterials-14-02034-f001:**
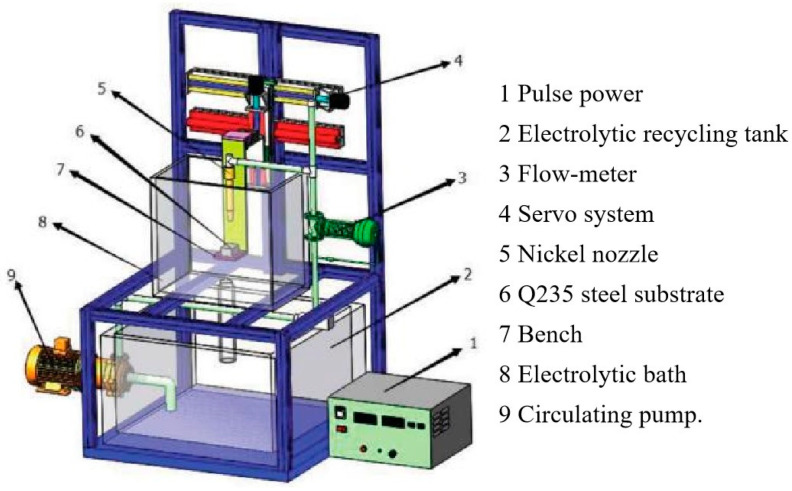
Schematic JPE system employed for manufacturing Ni/TiN composites.

**Figure 2 nanomaterials-14-02034-f002:**
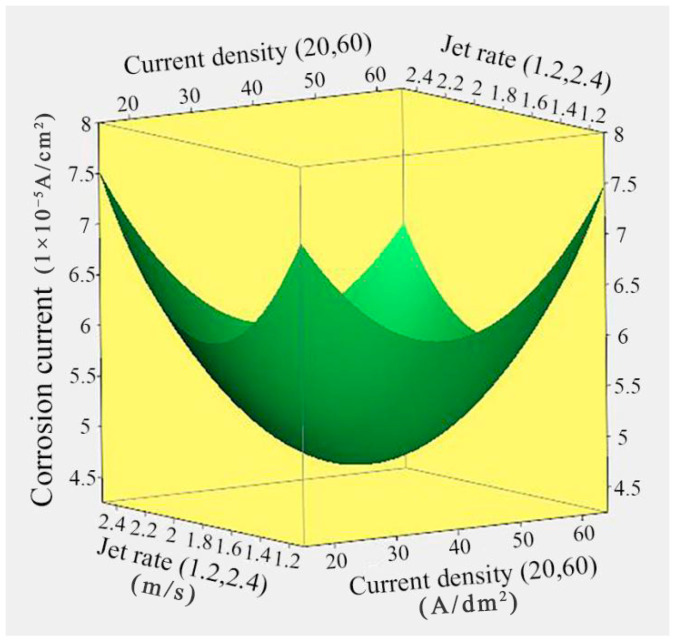
Response surface relationship between the corrosion current of Ni/TiN composites and the current density and jet rate.

**Figure 3 nanomaterials-14-02034-f003:**
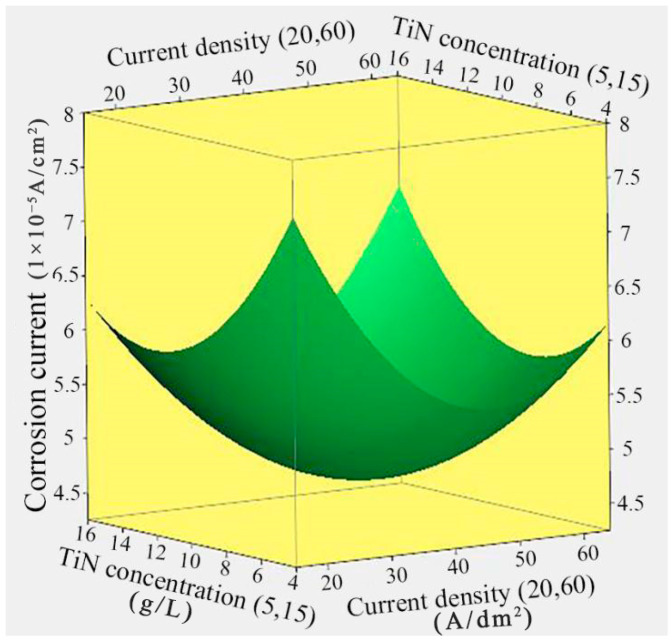
Response surface relationship between the corrosion current of Ni/TiN composites and the current density and TiN concentration.

**Figure 4 nanomaterials-14-02034-f004:**
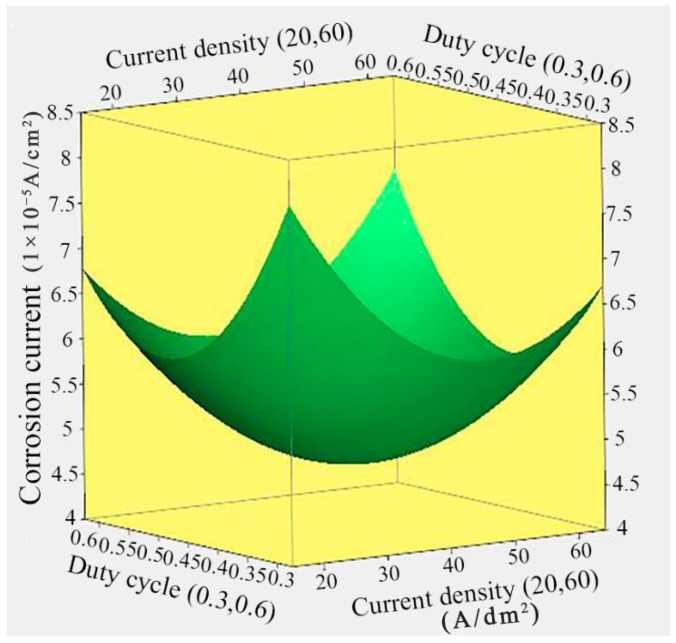
Response surface relationship between the corrosion current of Ni/TiN composites and the current density and duty cycle.

**Figure 5 nanomaterials-14-02034-f005:**
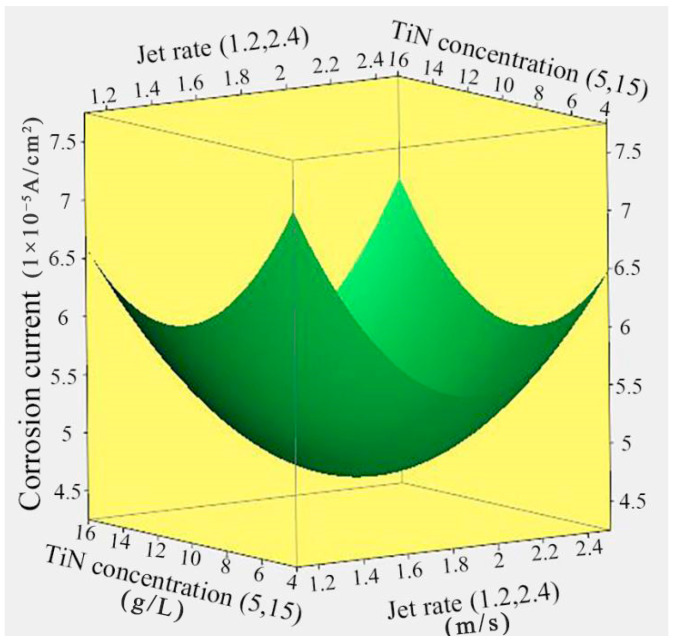
Response surface relationship between Ni/TiN composites and the jet rate and TiN concentration.

**Figure 6 nanomaterials-14-02034-f006:**
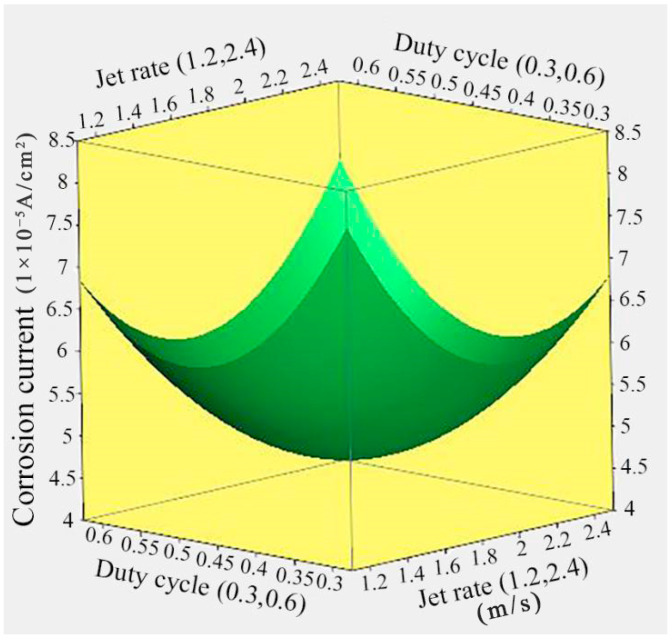
Response surface relationship between the corrosion current of Ni/TiN composites and the jet rate and duty cycle.

**Figure 7 nanomaterials-14-02034-f007:**
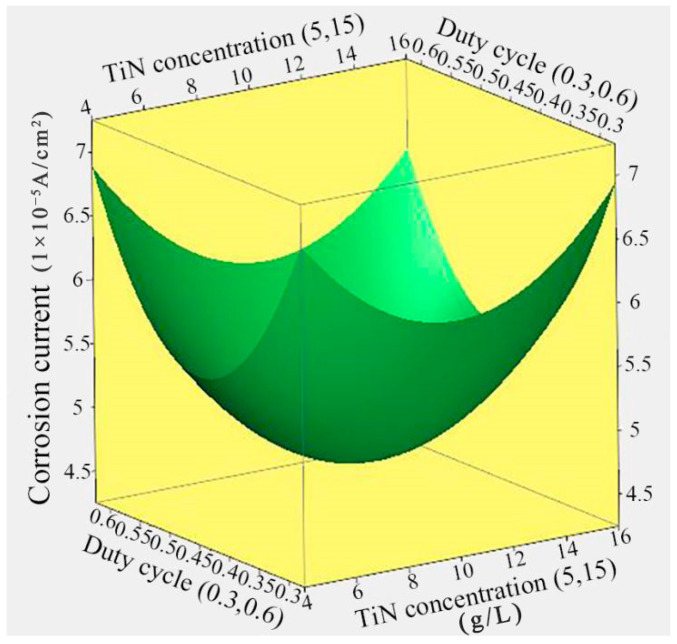
Response surface relationship between the corrosion current of Ni/TiN composites and the TiN concentration and duty cycle.

**Figure 8 nanomaterials-14-02034-f008:**
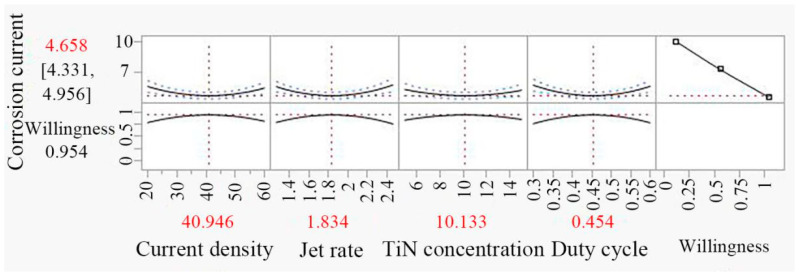
Operational parameter of Ni/TiN composites after optimization.

**Figure 9 nanomaterials-14-02034-f009:**
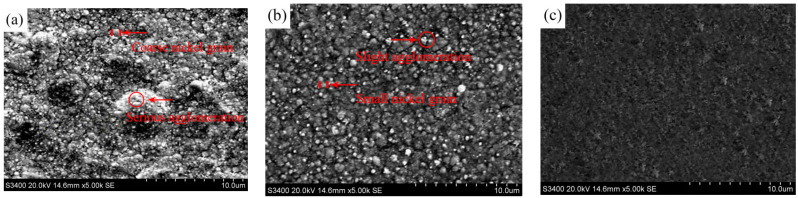
Surface morphological features of Ni/TiN composites: (**a**) JPE-1, (**b**) JPE-2, and (**c**) JPE-3.

**Figure 10 nanomaterials-14-02034-f010:**
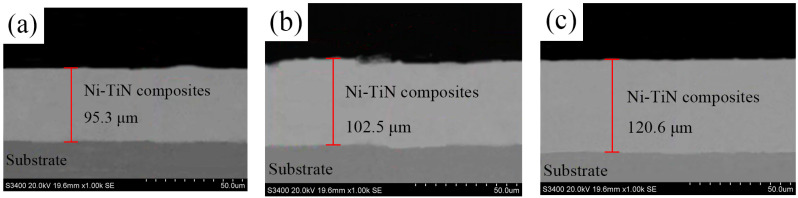
Cross-sectional SEM pictures of Ni/TiN composites: (**a**) JPE-1, (**b**) JPE-2, and (**c**) JPE-3.

**Figure 11 nanomaterials-14-02034-f011:**
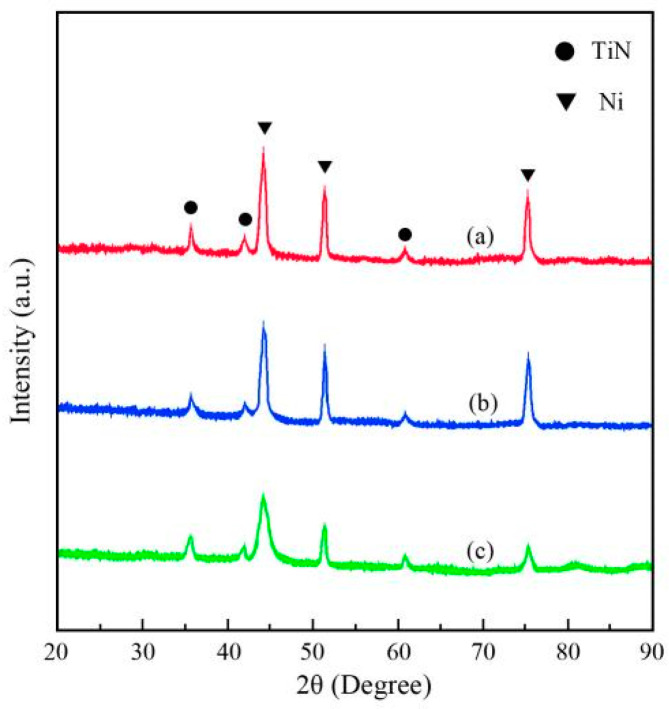
XRD patterns of Ni/TiN composites: (**a**) JPE-1, (**b**) JPE-2, and (**c**) JPE-3.

**Figure 12 nanomaterials-14-02034-f012:**
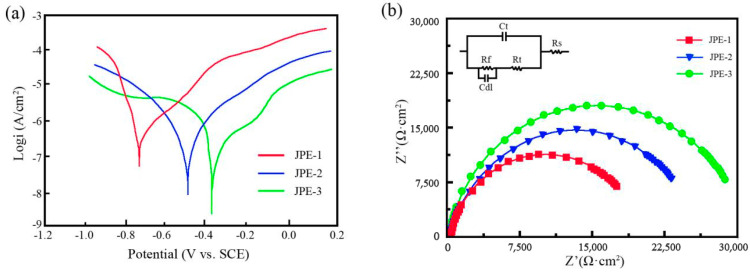
(**a**) Polarization curves and (**b**) Nyquist plots of Ni/TiN composites in 5 wt.% NaCl solution.

**Figure 13 nanomaterials-14-02034-f013:**
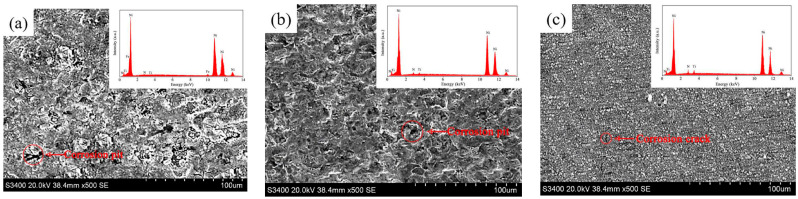
SEM graphs and EDS results of corroded surface of Ni/TiN composites: (**a**) JPE-1, (**b**) JPE-2, and (**c**) JPE-3.

**Table 1 nanomaterials-14-02034-t001:** Composition of electrolyte for prefabricating Ni/TiN composites.

Composition	Specific
NiSO_4_·6H_2_O	270 g/L
NiCl_2_·6H_2_O	35 g/L
H_3_BO_3_	20 g/L
C_6_H_8_O_7_	2.5 g/L
CTAB	60 mg/L
pH	4.3
Plating temperature	50 °C
Plating time	1 h

**Table 2 nanomaterials-14-02034-t002:** Pilot factors and level of CCD experiment.

Level	Pilot Factors			
	Current Density $x_{1}$ (A/dm^2^)	Jet Rate $x_{2}$ (m/s)	TiN Concentration $x_{3}$ (g/L)	Duty Cycle $x_{4}$
−1	20	1.2	5	0.30
0	40	1.8	10	0.45
1	60	2.4	15	0.60

**Table 3 nanomaterials-14-02034-t003:** Experimental results of corrosion current of Ni/TiN composites.

Number	$X_{1}$ (A/dm^2^)	$X_{2}$ (m/s)	$X_{3}$ (g/L)	$X_{4}$	Y (1 × 10^−5^ A/cm^2^)
1	20	1.2	5	0.6	7.61
2	20	2.4	15	0.3	8.22
3	20	1.2	15	0.6	6.86
4	20	1.2	5	0.3	9.52
5	20	2.4	15	0.6	7.79
6	20	1.2	15	0.3	7.91
7	20	2.4	5	0.3	8.28
8	20	1.8	10	0.45	5.87
9	20	2.4	5	0.6	8.33
10	40	1.8	10	0.45	4.71
11	40	1.8	10	0.45	4.71
12	40	1.8	10	0.6	5.37
13	40	2.4	10	0.45	5.36
14	40	1.2	10	0.45	5.89
15	40	1.8	5	0.45	5.58
16	40	1.8	15	0.45	4.87
17	40	1.8	10	0.3	5.89
18	60	1.2	5	0.6	7.89
19	60	2.4	15	0.6	8.43
20	60	1.2	15	0.3	8.45
21	60	1.8	10	0.45	5.08
22	60	1.2	5	0.3	8.35
23	60	2.4	5	0.3	6.06
24	60	2.4	15	0.3	7.92
25	60	1.2	15	0.6	8.34
26	60	2.4	5	0.6	8.33

**Table 4 nanomaterials-14-02034-t004:** Variance analysis of fitting model.

Source	Sum of Square	F-Ratio	*p*-Value
$X_{1}$	2.4538	17.4111	0.0016
$X_{2}$	0.8434	5.9841	0.0325
$X_{3}$	1.7382	12.3337	0.0049
$X_{4}$	2.1978	15.5945	0.0023

**Table 5 nanomaterials-14-02034-t005:** Element compositions and contents of Ni/TiN composites.

Ni/TiN Specimens	Ni Content (wt.%)	Ti Content (wt.%)	N Content (wt.%)
JPE-1	93.15	4.26	2.59
JPE-2	84.03	10.74	5.23
JPE-3	80.67	12.95	6.38

**Table 6 nanomaterials-14-02034-t006:** Average grain size of Ni/TiN composites.

Ni/TiN Specimens	Average Diameter of Ni Grain (μm)
JPE-1	0.64
JPE-2	0.35
JPE-3	0.12

**Table 7 nanomaterials-14-02034-t007:** Electrochemical corrosive data of Ni/TiN composites.

Ni/TiN Specimens	*E*_corr_ (V)	*I*_corr_ (1 × 10^−5^ A/cm^2^)
JPE-1	93.15	4.26
JPE-2	84.03	10.74
JPE-3	80.67	12.95

## Data Availability

Data are contained within the article.

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
