# Peer review of "Optimization for the Process Parameters of Nickel–Titanium Nitride Composites Fabricated via Jet Pulse Electrodeposition"

_nanomaterials, 2024, doi:10.3390/nano14242034_

Round 1
Reviewer 1 Report
Comments and Suggestions for Authors
The manuscript presents results that align with the scope of Nanomaterials. However, significant improvements are necessary. Specifically, the following aspects need attention: the use of precise scientific language and accurate terminology, inclusion of detailed experimental information, standardization of notation and units (e.g., duty cycle, JPED or JPE), and overall enhancement of the quality of the scientific language.
Specific recommendations:
1. Please use superscripts and subscripts in chemical formulas of ions and compounds, as well as for units and symbols in both the main text and references. Address issues such as missing spaces, punctuation (e.g., periods, commas), missing letters in authors’ names (e.g., ref. 4: "JarzÄ…bek"), incomplete volume and page numbers in references (e.g., ref. [6]), unnecessary dashes (e.g., refs. [22,23]) or spaces (e.g., refs. [12,27,29,32]), and capitalization errors (e.g., ref. [13]).
2. Line 32: The term “nickel-based coatings” can be interpreted in various ways, such as alloy coatings based on nickel. Consider using “nickel-matrix composites” to clearly convey that these are nickel-based materials containing ceramic particles as reinforcing phase.
3. Lines 34–35: The statement “NBCs are widely used in the transportation, mechanical, and processing industries” is too general. Provide specific applications in these fields.
4. Line 36: Cite specific references that describe the mentioned properties of nickel-matrix composite coatings.
5. Line 38: Briefly describe the JPED method in one or two sentences.
6. Lines 40–45: Rearrange this section to avoid repeating the name of the JPED method in every sentence. The context makes it clear what is being referred to.
7. Ensure consistent notation for composite materials. For instance, prefer Ni/SiC over Ni-Co-BN or vice versa, but apply one format throughout.
8. Line 49: Avoid vague terms like “performance.” Specify the property or behavior under discussion.
9. Line 51: I think you should replace “on the copper matrix” with “on the copper substrate”. The term “matrix” refers to a binder in composites, while “substrate” is the underlying material.
10. Lines 53–55: Revise the sentence “Li (…) content” for clarity.
11. Include more information about the properties of Ni/TiN coatings as well as main conclusions on the influence of JPED process parameters on the properties of these coatings based on previous works (cited).
12. Line 55 and following: The phrase “Previous literature” requires a reference. Consider omitting “Furthermore” and rewrite this paragraph starting a new section.
13. Line 77: Specify the rust and oil removers used.
14. Line 78: Clarify the term “20 sin-hydrochloric acid” (sin?).
15. Figure 1: Provide explanations for the figure elements marked as 1–9.
16. Table 1: Use correct formulas like NiSOâ‚„·6Hâ‚‚O (with a dot). Clarify whether anhydrous NiClâ‚‚ was used, or if it was NiClâ‚‚·6Hâ‚‚O or NiClâ‚‚·7Hâ‚‚O.
17. Line 87: Acknowledge that NiClâ‚‚ is also a source of nickel ions in the electrolyte.
18. Line 88: Specify what is meant by “inactivation of the anode”. Is it prevention against passivation? Define C₆H₈O₇ (citric acid?) and its concentration, as it is not mentioned in Table 1. Also, specify the bath pH.
19. Line 89: Clarify H₃BO₃’s function. It acts as a buffering agent in nickel plating electrolytes. What do authors mean by ‘a controlled release formulation’?
20. CTAB, as a cationic surfactant, provides a positive surface charge to ceramic particles, facilitating their transport towards negative charged electrode and incorporation into the metal layer on the cathode.
21. Specify TiN particle dimensions. Table 1 lists 8 g/L of TiN, but the discussion refers to 5–15 g/L. Reconcile this discrepancy.
22. Lines 96–98: Reorganize the sentence for better comprehension. Replace “to conduct the corrosion resistance” with “to optimize corrosion resistance”.
23. Line 108: Clarify what “duty” refers to—percentages of what?
24. Line 120: Indicate the context of CuKα and 20 kV.
25. Provide manufacturers’ details for the experimental equipment.
26. Specify details of corrosion measurement procedures.
27. Figures 2–7: Include units for all figures.
28. Lines 134–135: Elaborate on how increased current density influences TiN particle co-deposition and nickel ion deposition.
29. Confirm your interpretation of concentration polarization and clarify if powdery deposits were observed. How does ref. [21] on zinc and ref. [31] relate to Ni/TiN composites?
30. Lines 139–140: Explain “point discharge”.
31. Line 141: Address why the jet rate influences particle size refinement. Provide evidence.
32. Lines 144–145: Show evidence for the claim that high jet rates affected cathode surface (?) inhibiting coating growth and increasing corrosion current of composite coatings.
33. Lines 162–163: Explain “concentration polarization” as depending on TiN concentration and provide evidence for high electrolyte viscosity.
34. The results presented in Chapters 3.1–3.6 were discussed based on other studies (references), but the discussion relies on very general statements with no clear evidence or detailed explanations. This represents a significant weakness in the paper. Strengthen the discussion by providing deeper insights. Discuss the relevance of reporting parameter values to the hundredths decimal place.
35. Figure 11: Compare relative intensities of XRD peaks, not intensities “as-received”.
36. Include and explain the Scherrer formula to estimate nickel matrix grain size. Discuss differences among sizes of crystallites, grains, and surface nodules, ensuring alignment between text and Table 6.
37. Describe how corrosion potentials were identified. The corrosion potential is not the minimum potential at the transition between the cathodic and anodic polarization curves. Specify sample immersion times in NaCl and emphasize the scientific aspects of corrosion studies.
38. Ensure consistency between TiN contents reported in conclusions and Ti contents shown in the results (these are not the same values!). Discuss the basis for estimating Ti content from surface analysis and the absence of particle distribution data across the coating thickness.
39. Use explicit conditions for JPE-1 to JPE-3 to avoid requiring readers to search the text.
40. Avoid abbreviations in the abstract unless they are well-known and clearly defined. Rewrite the abstract to present significant results clearly and avoid generalities.
Comments on the Quality of English Language
Improvement the quality of scientific languag by using correct vocabulary and terminology, as well as avoiding generalities are recommended.
Reviewer 2 Report
Comments and Suggestions for Authors
In their systematic study titled “Optimization for the Process Parameters of Nickel-Titanium Nitride Composites Fabricated via Jet Pulse Electrodeposition,” Menguy Cao et al. explored key fabrication parameters of anti-corrosion coatings and their interrelationships. Using Response Surface Methodology (RSM), they identified optimal conditions for achieving superior corrosion resistance, with an observed-to-predicted corrosion current error of only 0.64%. Additionally, they provided detailed characterization of three anti-corrosive composites, demonstrating that TiN content plays the most important role in minimizing corrosion currents. After reviewing their work, I suggest that the article requires major revision to be suitable for publication in Nanomaterials.
The article is within the scope of the journal, title is satisfactory, abstract covers the pertinent points. The article is of fair originality and significance in the field, and possesses solid scientific quality. Figures are presented in decent fashion, and the conclusions are supported by the data, but they need to be presented in the narrative form, and not in bullets.
List of suggested changes, questions, and comments to be addressed by the authors:
Line18: However is twice written. The authors are kindly asked to carefully grammar check the article, and correct all typos.
Line 60: Please support the statement with an appropriate reference. “The response surface method (RSM)…"
Numbers in formulas are presented as normal text, whereas some should be in subscript, and some should be in superscript.
In Table 1, the units should be presented along with the numerical values for better clarity.
Line 145: The authors are kindly ask to provide more details about consistency between their work and the work presented in reference 23.
Line 250: The image of the sample JPE-3 has low contrast and brightness, so it is hard to conclude anything about nickel grains and distribution of the TiN nanoparticles. The authors are kindly ask to rephrase the corresponding conclusion.
Line 254: It remains unclear how the presented results and conclusions can be related to the reference 32, which reports on the Ni-La2O3 nanocomposites. The authors are kindly asked to elaborate this.
Line 295: The authors are kindly asked to present more context regarding the chosen equivalent circuit and the impedimetric measurements. Parameters for the EIS measurements should be disclosed in the Experimental section.
Conclusions should be presented in narrative form.
Round 2
Reviewer 1 Report
Comments and Suggestions for Authors
The quality of the article has been improved in accordance with the review. However, it still requires minor editorial adjustments, which can be made during the final editing phase and do not need to be reviewed by the referee. The authors should bear in mind that they are publishing a scientific article, so it is essential to maintain an appropriate standard of writing.
- Line 37: What is “oil-gas X52 steel”? Please improve this term to clarify its meaning.
- Notation of composites: Please verify whether the notation for composites is used correctly throughout the text. Note that the symbol “/” denotes a phase boundary between composite components. If the composite matrix is an alloy, use the symbol “-” to separate alloy components, e.g., Ni-W/TiN (Ni-W alloy matrix containing TiN particles) instead of Ni/W/TiN (Ni matrix containing W particles and TiN particles); or Ni-Co/SiO2 instead of Ni/Co/SiO2, etc. Please review the entire text for consistency. Correct notation is crucial for accurately interpreting the material type.
- Lines 46-47: When adding revisions, ensure they align with the context of the text as a whole. Please note that the newly added sentence: (1) repeats the following sentence regarding current efficiency, and (2) does not clearly specify what “improved quality” refers to.
- ‘Matrix’ terminology: Please verify whether the term ‘matrix’ is used correctly throughout the text. For example, in line 51, it appears to be used to mean ‘substrate.’
- Table 1: Please organize the compounds by grouping them together, followed by pH values and process parameters.
- Line 144: What does “door temperature for 10 min” mean? Please clarify.
- There is a codeposition of nickel (as metal, not ions) with particles: lines 187 and 205 .
Author Response
Thank you for your advise. According to your suggestion, we have revised in the manuscript. We will also pay attention to the issues you raised in our future paper writing.

Reviewer 2 Report
Comments and Suggestions for Authors
The authors have successfully addressed all the changes suggested by me in the revision, and have answered the questions that I had. The manuscript can be accepted for publication in the present form.
Author Response
Thank you for your recognition. We look forward to receiving your guidance in our future work